# Epidemiology of injured patients in rural Uganda: A prospective trauma registry's first 1000 days

**Dennis J. Zheng**[1], **Patrick J. Sur**[2], **Mary Goretty Ariokot**[3], **Catherine Juillard**[4], **Mary Margaret Ajiko**[3], **Rochelle A. Dicker**[4] *

**1** Department of Surgery, University of California Los Angeles, Los Angeles, California, United States of America, **2** UC Riverside School of Medicine, University of California, Riverside, California, United States of America, **3** Department of Surgery, Soroti Regional Referral Hospital, Soroti, Uganda, **4** Department of Surgery, Program for the Advancement of Surgical Equity, University of California, Los Angeles, California, United States of America

* rdicker@mednet.ucla.edu

**Data Availability Statement:** All relevant data are within the manuscript and its Supporting Information files.

## Abstract

Trauma is a leading cause of morbidity and mortality worldwide. Data characterizing the burden of injury in rural Uganda is limited. Hospital-based trauma registries are a critical tool in illustrating injury patterns and clinical outcomes. This study aims to characterize the traumatic injuries presenting to Soroti Regional Referral Hospital (SRRH) in order to identify opportunities for quality improvement and policy development. From October 2016 to July 2019, we prospectively captured data on injured patients using a locally designed, context-relevant trauma registry instrument. Information regarding patient demographics, injury characteristics, clinical information, and treatment outcomes were recorded. Descriptive, bivariate, and multivariate statistical analyses were conducted. A total of 4109 injured patients were treated during the study period. Median age was 26 years and 63% were male. Students (33%) and peasant farmers (31%) were the most affected occupations. Falls (36%) and road traffic injuries (RTIs, 35%) were the leading causes of injury. Nearly two-thirds of RTIs were motorcycle-related and only 16% involved a pedestrian. Over half (53%) of all patients had a fracture or a sprain. Suffering a burn or a head injury were significant predictors of mortality. The number of trauma patients enrolled in the study declined by five-fold when comparing the final six months and initial six months of the study. Implementation of a context-appropriate trauma registry in a resource-constrained setting is feasible. In rural Uganda, there is a significant need for injury prevention efforts to protect vulnerable populations such as children and women from trauma on roads and in the home. Orthopedic and neurosurgical care are important targets for the strengthening of health systems. The comprehensive data provided by a trauma registry will continue to inform such efforts and provide a way to monitor their progress moving forward.

**Funding:** Dennis J. Zheng received a Doris Duke International Clinical Research Fellowship from the Doris Duke Charitable Foundation. The funders had no role in study design, data collection and analysis, decision to publish, or preparation of the manuscript.

**Competing interests:** The authors have declared that no competing interests exist.

## Introduction

Traumatic injury is a leading cause of morbidity and mortality worldwide, accounting for nearly four and a half million lives lost per year, or approximately 8% of the world's annual deaths [1]. Low- and middle-income countries (LMICs) bear this weight disproportionately, with roughly 90% of injury-related deaths occurring there [2]. Injuries are likely to grow even more common over the coming decades due to continued urbanization and infrastructural development [3]. Despite their heavy burden of injury, trauma care in LMICs is plagued by major deficiencies in capacity and access [4].

The process of improving trauma care in such locations is hindered by a lack of primary data, which limits the development of interventions to address the problem of injury. Though helpful, hospital medical records rarely contain the full range of information necessary to prioritize and evaluate efforts surrounding trauma care. Access to comprehensive, context-specific data regarding patterns of injury and subsequent care delivery is vital for all parties involved in a trauma system, including clinicians, health researchers, and policymakers.

In recognition of this need, hospital-based trauma registries have been well-studied as effective methods of measuring burden of injury, guiding improvement efforts, and assessing ongoing quality of care, especially in high-income country settings [5]. While trauma registries in LMICs have historically been less common, over the past two decades several have been implemented and analyzed in Uganda, where injury is a significant cause of death and disability. In fact, Uganda ranked 4th out of 15 eastern Sub-Saharan African countries in most disability-adjusted life years (DALYs) due to road traffic injury in 2017 [6]. The heavy burden of injury in the urbanized Ugandan capital of Kampala has been documented extensively [7]. However, there is a dearth of information available on the patterns of injury in the more rural portions of the country, where over 80% of the country's residents live [8].

To assemble a more complete data set than previously available regarding traumatic injury in Uganda, a prospective trauma registry was established at a regional referral hospital in eastern Uganda beginning in 2016 through collaboration among Ugandan and American partners. This study aimed to evaluate the patterns of demographics, injury characteristics, clinical markers, and health outcomes of patients arriving at Soroti Regional Referral Hospital (SRRH) for trauma care during the first 1000 days of the registry's existence.

## Methods

### Study setting

Data were collected prospectively between October 2016 and July 2019 at SRRH, one of 13 public regional hospitals in Uganda. The government-run 250-bed facility serves a predominantly rural catchment population of two million people, or roughly 5% of the Ugandan population. On an annual basis, 21,000 inpatients and 103,000 outpatients receive care at the hospital, with a referral base of eight district level hospitals throughout the region [9]. Representing the second-highest level of care within the national health system, SRRH offers patients their first opportunity to access specialized surgical care at any time of day or night. Patients arriving at the hospital, which lacks a dedicated casualty department, may be initially evaluated in its general outpatient or orthopedic clinics or, depending on injury severity, may be admitted directly to hospital wards.

### Data collection

A structured questionnaire was developed for the registry based on hospital focus group discussions, an initial pilot study at SRRH, and analysis of other LMIC trauma registries as well as

WHO trauma guidelines. Months prior to the initiation of this study, multiple groups of hospital surgeons, intern doctors, and clinical officers had been surveyed regarding what types of injury information they wished to collect and how to best incorporate the registry into their work. A month-long pilot study involving roughly 50 practitioners across the hospital was held and demonstrated the feasibility of detailed data collection balanced against the need to maintain efficient clinical workflow. Based on these discussions and findings, a revised single-page registry form (S1 Appendix) was developed for the larger study. The form was first used at the time of initial patient encounter to collect data on demographics, pre-hospital care, details of injury, and preliminary clinical assessment, in addition to vital signs (blood pressure, pulse, respiratory rate, AVPU neurological status) on presentation. Due to concerns about the feasibility of collecting anatomical injury information, the Kampala Trauma Score (KTS) was chosen to categorize severity of injury (KTS 14–16 signifying a mild injury, 11–13 a moderate injury, 10 or below a severe injury). This physiologic-based scoring system has been validated as a useful tool for predicting mortality especially in LMIC settings [10]. The form also included items of particular interest in rural Uganda, pertaining to helmet use in road traffic injury or involvement of mob justice. Patients admitted to a SRRH ward were followed up through their eventual disposition from the hospital (e.g., discharge home, transfer to the national hospital in Kampala, or death). Following multiple training sessions conducted over several weeks, forms were completed by local clinicians and trained research assistants, who then transferred data to an electronic database in REDCap hosted at University of California, San Francisco [11]. REDCap is a secure, web-based application designed to support data capture for research studies. Inclusion criteria included any injured patient presenting to hospital for initial evaluation of injury or any patient referred from a district hospital for injury evaluation. Isolated soft tissue injuries were excluded.

## Data analysis

Analyses were performed using Stata, version 15.1 [12]. Descriptive statistics and tabulations were generated on demographic characteristics, mechanism of injury, prehospital care, body region injured, and type of injury, as well as clinical outcomes. Chi-square tests were applied to identify associations between demographic characteristics and nature of injury. Univariate and multivariate logistic regression models were developed to characterize identified associations between nature of injury and clinical outcomes. An alpha value of 0.05 was used as a threshold for statistical significance in our analyses.

## Ethical approval

Written consent could not be feasibly obtained for all study patients due to the acuity of their medical injuries. Oral informed consent was granted by all adult patients during their hospital encounter, with permission from parents/guardians obtained for all patients under 18 years of age, and documented in the registry. This study protocol was approved by the Mulago National Referral Hospital Research Committee, the Uganda National Council of Science and Technology, and the Institutional Review Board of the University of California, San Francisco.

## Results

### Demographic characteristics

A total of 4109 patients were entered into the trauma registry during the 33-month study period. Roughly two-thirds (62.7%) of the study population were male, and the median age was 26 (IQR 9–37) years. Patients ranged from 1 month to 107 years of age, with 25% of

patients ranging 0–9 years. Most patients (61%) were residents of Soroti district. The leading occupations of injured patients were student (32.7%) or peasant farmer (30.8%) (Table 1).

## Nature of injury

The most common mechanisms of injury were falls (35.7%) and road traffic injuries (RTIs; 34.9%). Penetrating mechanisms including stabbing/cuts, animal bites, and gunshot wounds were relatively infrequent (6.8%). Falls were significantly more common among women ($p < 0.001$) and patients age 0–19 ($p < 0.001$), while RTIs were significantly more common among men ($p < 0.001$) (Fig 1). Most injuries occurred at home (46.8%) or on roads/streets (40.9%), and they were predominantly unintentional (81.9%). Mob justice was a factor in just 1% of cases (Table 2).

Among patients injured in an RTI, the majority were either passengers or drivers/riders of a motor vehicle (83.6%). Almost two-thirds (62.7%) of RTIs involved a motorcycle, and 13.4%

**Table 1. Demographics of injured patients at SRRH.**

| Sex | Frequency | Percent |
|---|---|---|
| Male | 2,559 | 63% |
| Female | 1,529 | 37% |
| Total | 4,088 | 100% |
| **Age** | | |
| 0–9 | 1,028 | 25% |
| 10–19 | 734 | 18% |
| 20–29 | 823 | 20% |
| 30–39 | 573 | 14% |
| 40–49 | 327 | 8% |
| 50–59 | 217 | 5% |
| 60–69 | 407 | 10% |
| 70–79 | 122 | 3% |
| 80–89 | 74 | 2% |
| 90–99 | 14 | <1% |
| 100–109 | 2 | <1% |
| Total | 4,109 | 100% |
| **Home district** | | |
| Soroti | 2,437 | 61% |
| Amuria | 570 | 14% |
| Serere | 394 | 10% |
| Kaberamaido | 190 | 5% |
| Katakwi | 194 | 5% |
| Ngora | 106 | 3% |
| Kumi | 18 | <1% |
| Bukedea | 6 | <1% |
| Other | 84 | 2% |
| Total | 3,999 | 100% |
| **Occupation** | | |
| Peasant farmer | 1,479 | 37% |
| Student | 1,242 | 31% |
| Unemployed | 629 | 16% |
| Other | 658 | 16% |
| Total | 4,008 | 100% |

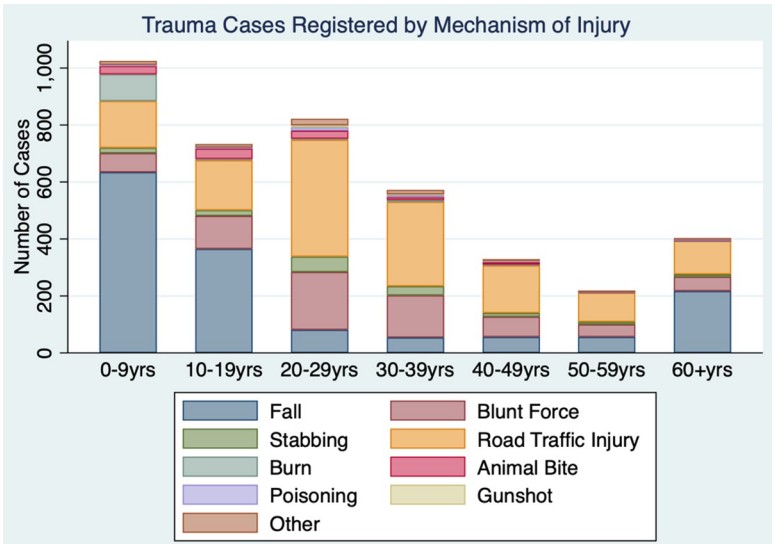

**Fig 1. Mechanism of injury by patient age group at SRRH.**

of RTIs were confirmed or suspected to involve intoxication with alcohol or other substances. In only 5.4% of motorcycle or bicycle-related RTIs had a helmet been worn by the victim of injury (Table 2).

## Mode of arrival and time to encountering care

More than three-fourths (76.2%) of patients arrived at the hospital using motorcycle, while only 7.2% had been transported via ambulance. One-fifth (20.0%) of all patients had been referred from another health care facility prior to arrival. Those arriving by ambulance were significantly more likely to have a moderate or severe injury compared to those arriving via other means ($p < 0.001$). The median duration of time from time of injury to arrival at hospital was 7.5 (IQR 21.3) hours, with only 18.9% arriving within one hour and 45.8% arriving within six hours after suffering their injury. Of those traveling for greater than six hours, 43% were residents of a non-Soroti district (Table 3).

## Clinical characteristics

The most common anatomic locations of injury were the upper (37.7%) and lower extremities (35.8%), followed by the head/neck (26.2%). The most common types of injury were fracture (30.9%), bruise/abrasion (21.8%), head injury (15.8%) or laceration/bite (13.4%). Upon arrival, initial systolic blood pressure was recorded for 58.9% of patients; initial pulse rate was recorded for 66.3%; initial respiratory rate was recorded for 63.5%; and initial neurological status was recorded for 95.3%. Sufficient data were available to calculate a KTS value for 52.1% of patients, and of those 16.8% were moderately or severely injured. Of the severely injured patients, 73.3% had been involved in RTIs (Table 4).

## Outcomes

Upon arrival at the hospital, 40.9% of trauma patients were treated and sent home, 58.9% were admitted to inpatient wards, and 1.4% were transferred or referred elsewhere. Of those admitted, 1268 patients (30.9%) underwent a procedure or operation as an inpatient. A total of 66 trauma patients (1.6%) died upon initial presentation or after hospital admission. Results of

**Table 2. Injury characteristics of patients at SRRH.**

| Injury setting | Frequency | Percent |
|---|---|---|
| Home | 1,901 | 47% |
| Road/Street | 1,661 | 41% |
| Work | 78 | 2% |
| School | 204 | 5% |
| Other | 216 | 5% |
| Total | 3,999 | 100% |
| **Intent** | Frequency | Percent |
| Unintentional | 3,304 | 82% |
| Intentional, assault | 662 | 16% |
| Intentional, self | 25 | 1% |
| Mob justice | 26 | 1% |
| Unknown | 20 | 1% |
| Total | 4,008 | 100% |
| **Vehicle type** | Frequency | Percent |
| Vehicle | 329 | 23% |
| Motorcycle | 880 | 61% |
| Bicycle | 195 | 14% |
| Unknown | 28 | 2% |
| Total | 1,432 | 100% |
| **Helmet use** | Frequency | Percent |
| Yes | 42 | 5% |
| No | 459 | 78% |
| N/A | 165 | 17% |
| Total | 967 | 100% |
| **Role in incident** | Frequency | Percent |
| Driver or rider | 498 | 35% |
| Passenger | 647 | 45% |
| Pedestrian | 224 | 16% |
| Unknown | 63 | 4% |
| Total | 1,432 | 100% |

**Table 3. Pre-hospital transportation and delays to presentation of patients arriving to SRRH.**

| Mode of arrival | Frequency | Percent |
|---|---|---|
| Motorcycle or Taxi | 3,135 | 76% |
| By foot | 154 | 4% |
| Private Car/Bicycle | 373 | 9% |
| Ambulance/Police | 297 | 7% |
| Unknown | 150 | 4% |
| Total | 4,109 | 100% |
| **Time from injury to hospital arrival** | Frequency | Percent |
| Less than 1 hour | 484 | 12% |
| 1 to 6 hours | 1,359 | 33% |
| Over 6 hours | 1,974 | 48% |
| Unknown | 292 | 7% |
| Total | 4,109 | 100% |

**Table 4. Clinical characteristics of injured patients at SRRH.**

| Injury location | Frequency | Percent |
|---|---|---|
| Upper Extremity | 1,522 | 38% |
| Lower Extremity | 1,468 | 36% |
| Head or Neck | 1,083 | 26% |
| Face | 529 | 13% |
| Chest | 408 | 10% |
| Abdomen, Pelvis, or Perineum | 379 | 9% |
| Back or Spinal Cord | 207 | 5% |
| Total | 5,626 | |
| **Diagnosis** | Frequency | Percent |
| Fracture | 1,936 | 47% |
| Bruise or Abrasion | 1323 | 32% |
| Head Injury | 857 | 21% |
| Laceration or Bite | 786 | 19% |
| Sprain/Dislocation | 233 | 6% |
| Abdominal Injury | 206 | 5% |
| Thoracic Injury | 161 | 4% |
| Burn | 117 | 3% |
| Other Injury | 55 | 1% |
| Spinal Cord Injury | 18 | <1% |
| Total | 5,674 | |
| **Injury severity** | Frequency | Percent |
| Mild Injury | 1,781 | 43% |
| Moderate Injury | 343 | 8% |
| Severe Injury | 15 | <1% |
| Unknown | 1,970 | 48% |
| Total | 4,109 | 100% |

multivariate logistic regression displayed that moderate or severe injury (OR = 8.34, 3.24–21.5 95% CI), burn as mechanism of injury (OR = 6.67, 2.69–17 95% CI), and presence of head injury (OR = 3.48, 2.04–5.96 95% CI) were independent significant predictors of mortality after controlling for age, sex, and referral status (Table 5).

## Data completeness

During the first six months of the study period, the registry recorded an average of 265 patients per month, for a total 1592 patients (38.7% of the entire study population). Rates of data capture over months 7–12 were 134 patients per month, or 805 total (19.6% of the total population). The registry recorded 102 patients per month, or 1224 total (29.8% of the total population) in months 13–24, and 54 patients per month, or 488 total (11.9% of the total population) in months 25–34 (Table 6).

## Discussion

This study demonstrated the significant burden of trauma in patients seeking care at SRRH, as well as the effectiveness of a prospective trauma registry in documenting trends of injury over time. Primary data regarding the causes of trauma, its associated risk factors, and subsequent therapies are vital components in the strengthening of health systems in LMIC settings [13]. The detailed examination of local epidemiology provided by this study highlights the potential

**Table 5. Multivariate logistic regression of significant predictors of mortality of injured patients at SRRH.**

| Predictors | Crude odds of subsequent mortality (95% CI) | Adjusted odds of subsequent mortality (95% CI) | P-value |
|---|---|---|---|
| Age >18 | 1.02 (1.01–1.03) | 1.03 (1.02–1.04) | <0.001 |
| Male sex | 1.22 (0.721–2.07) | 1.3 (0.77–2.2) | 0.458 |
| Referred from elsewhere | 1.76 (1.01–3.05) | 1.7 (0.977–2.94) | 0.045 |
| Moderate or severe injury | 8.76 (3.42–22.4) | 8.34 (3.24–21.5) | <0.001 |
| Burn mechanism of injury | 3.35 (1.48–8.29) | 6.67 (2.69–17) | <0.001 |
| Head injury | 3.51 (2.14–5.76) | 3.48 (2.04–5.96) | <0.001 |

*Adjusted for age, sex, referral status, mechanism of injury, type of injury.

of hospital-based registries to guide injury prevention and quality improvement efforts throughout sub-Saharan Africa.

Echoing the findings of other LMIC trauma registries, injured patients presenting to SRRH tended to be male. The sex imbalances in those affected by trauma are likely related to specific occupations and recreational activities, underlining the broader economic and social implications of injury. For example, men were overrepresented in work-related injuries, whereas women were proportionally more likely to have suffered a burn or animal bite. Neither sex was spared by road traffic injury, however. Sub-Saharan lays claim to some of the highest rates of transportation-related and fatalities in the world, reaching up to 65 deaths per 100,000 on some estimates [14]. While not dissimilar in its findings, this study was able to demonstrate the particular risks borne by riders of motorcycles ('boda bodas') and their passengers, the vast majority of whom were not wearing helmets. Nearly 15% of incidents were suspected or confirmed to involve substance intoxication. The substantial rates of injury associated with boda bodas in Uganda are well-documented, due in part to the country's poorly enforced speed limits and traffic safety laws [15]. Only 15% of patients injured on the road in Soroti were pedestrians–a stark contrast from other studies in Uganda and neighboring countries, perhaps attributable to the rural nature of the hospital's catchment area [16, 17]. Our findings reinforce the importance of directing public safety interventions towards two-wheeled modes of transportation in LMICs, where poor road conditions and lack of safe transportation alternatives remain major challenges [18].

Patients under the age of 10 made up roughly one-fourth of the study's population. This suggests an immense burden of pediatric injury in Uganda, where 48% of people are between the age of 0–14 –the second-highest proportion in the world [19]. Falls in particular were a problematic mechanism for children, who are known to often tumble from towering fruit trees in the region [20]. A comparison with trauma registries in nearby countries reveals

**Table 6. Trauma registry enrollment over study period.**

| Time period | Individuals enrolled | Percent |
|---|---|---|
| July 2016 to Dec 2016 | 1,007 | 25% |
| Jan 2017 to June 2017 | 965 | 23% |
| July 2017 to Dec 2017 | 745 | 18% |
| Jan 2018 to June 2018 | 679 | 17% |
| July 2018 to Dec 2018 | 437 | 11% |
| Jan 2019 to June 2019 | 238 | 6% |
| Unknown | 38 | <1% |
| Total | 4,109 | 100% |

exceedingly similar patterns of injury in pediatric populations in Malawi, Kenya, and others [21, 22]. Public safety programs in resource-poor settings should be targeted towards youth, especially residents of non-urban areas, to prioritize the topic of fall prevention. Advancements in local food security and poverty relief would also likely exert a downstream effect in this area. An additional mechanism of injury crucial for further investment is burn injury, which was found to be a significant predictor of mortality in all age groups in our study. Strategies to reduce the incidence of burns might include provision of safe cooking, heating, and lighting devices in the home and education campaigns for patient caregivers in non-urban communities [23].

Head injury was the clinical diagnosis most associated with death in the study population. Contributing factors in SRRH include associated injury severity, the absence of diagnostic capabilities beyond X-ray, and a lack of local neurosurgical specialists. Patients suffering any significant injury to the brain or spinal cord in Soroti are typically transferred to Mulago Hospital, the national tertiary care center located more than six hours away, where mortality rates associated with traumatic brain injury have been found to reach up to 55% [24]. A recent review of trauma registry data generated across five hospitals in the neighboring country of Tanzania, where access to specialized neurosurgical care is similarly restricted to a single national center, noted that intracranial injury accounted for over 80% of all trauma-related deaths [25]. The authors recommend further research regarding pre-hospital and hospital interventions aimed at stabilizing patients with head injury in low-resource settings.

While not a significant contributor to mortality in the study, the morbidity of bony and soft tissue injuries was substantial. Over half of all trauma patients in the SRRH registry were found to have either a fracture or sprain, signaling a burden of orthopedic injury dwarfing available resources for care. Inpatient X-rays are not consistently available, and due to infrastructural limitations, patients presenting with fractures in need of operative repair are referred elsewhere for surgical management. Efforts to improve the diagnostic and therapeutic options available to patients with orthopedic injuries in Soroti and similar settings would be widely beneficial. The national economic impact of long-bone fractures in Uganda has been estimated to be at least 32 million USD per year [26].

Registry data also offers helpful information regarding the ways in which local patients seek and access emergency care. Most patients traveled to the hospital by private means, using hired motorcycles or automobiles, with far fewer utilizing ambulances for transport to the hospital. Critically, over 80% of patients were unable to reach the hospital within an hour of sustaining their injuries. Improvements in pre-hospital care to reduce delays from time of injury to access to care, including increased provision of ambulances and facilitation of the referral process from community health centers, would benefit trauma patients across the region.

Finally, it is possible that a trauma registry may indirectly affect the practice of trauma care. An analysis of randomly sampled patient charts from the year prior to registry implementation revealed that the percentage of patients with a documented set of vital signs increased significantly after implementation of the registry. The Hawthorne effect, a phenomenon in which the behavior of subjects changes because they know they are being studied or observed, has been previously described as a potential source of bias in surgical research [27]. It is plausible that increased awareness of the trauma registry and the requirements of its associated paper form may have led staff to feel more compelled to obtain and record vital signs for their patients. Similarly, authors from Botswana noted a significant uptick in the rates of primary and secondary survey completion in trauma patients after implementation of a pilot trauma registry; whether this was due to improvement in clinical practice or simply adherence to documentation is unclear [28]. The SRRH registry also collects data on time of patient arrival at the hospital and time of initial clinical encounter, which allows for detailed, hour-by-hour

understanding of trauma patient presentation patterns. Analyses of this data can be utilized for improved physical and human resource allocation for trauma care by hospital administration or government health officials.

Over the initial 1000 days of registry data collection, the quantity of patients entered into the trauma registry on a regular basis steadily decreased until the number of newly injured individuals added each month was less than half of what it had been prior. Rather than a success of injury prevention efforts, it is likely this represented an increasing number of patients who eluded inclusion into the registry. The decline in data completeness demonstrates the difficulty of continuously maintaining a trauma registry in an LMIC setting, where consistently high patient volumes and a lack of medical documentation infrastructure already impede the provision of clinical care. Trauma registries can be valuable sources of information when compared to municipal or administrative databases [29]. Building them up via thorough data collection in this type of environment requires an array of individual and systemic factors to work in concert to overcome the burden of having too many patients and not enough paper. Over the course of the study period, the responsibility of day-to-day registry management was transitioned to local research staff, who were able to be present in the hospital during daytime hours only. Due to the limited medical charting at SRRH, it was difficult to retrospectively gather information on patients who had presented overnight, as clinicians were too busy to prioritize the documentation process. Workforce turnover and shortages remained commonplace, meaning that awareness of the registry's workings varied a great deal among hospital staff over time. Other factors such as the lack of a dedicated casualty department to provide centralized triage of patients rendered the task of tracking patients as they moved throughout the hospital challenging. Nonetheless, the ingredients for successful trauma registry implementation that have been well-documented in the literature, including local stakeholder buy-in, a motivated workforce, and secure funding, were crucial elements in the development, implementation, and ongoing presence of the SRRH registry. One way to address its shortcomings might be to transition away from the use of paper, as numerous studies have noted the high rates of data completeness and correctness associated with electronic trauma registries [30]. Unfortunately, their implementation is not yet feasible in a setting like rural Uganda.

This study has several limitations. A single hospital-based trauma registry is susceptible to potential selection bias in that the sample population is limited to injured persons seeking care at the facility. The low utilization of formal medical services in developing countries is likely to underestimate the number of injured patients, and thus our findings may not be representative of the true burden of injury in Soroti. Further exploration of care-seeking behaviors of the injured is likely to yield valuable insight in this population, as it has elsewhere [31]. Similarly, the injury-related mortality rate of approximately 16 deaths per 1000 patients in our study is likely an underestimate, as pre-hospital deaths were unable to be captured.

## Conclusions

Our study shows that a trauma registry can be a useful source of data for quantifying the burden of injuries and patient outcomes in a Ugandan regional referral hospital. Comparing and contrasting our registry's findings to those of neighboring registries in sub-Saharan Africa sheds further light on the challenges faced in rural LMIC trauma care. Vulnerable populations such as children would benefit greatly from efforts to improve road safety and burn prevention, while expanding access to quality pre-hospital, neurosurgical, and orthopedic care are important targets for the strengthening of health systems. Finally, the difficulty associated with maintaining reliable data collection reinforces the need for a centralized casualty department to serve patients entering the hospital. All of these lessons will only become more salient

during the development of a national surgical, obstetric, and anesthesia plan (NSOAP) in Uganda. Continued collection and analysis of registry data will facilitate the design of clinical quality improvement initiatives and implementation of public policy changes to decrease the substantial burden of injury in Soroti and beyond.

## Supporting information

**S1 Table. The registry data set used for analysis.**
(XLSX)

**S1 Appendix. The trauma registry surveillance form used for data collection.**
(PDF)

## Acknowledgments

We wish to acknowledge the contributions of the staff and patients at Soroti Regional Referral Hospital, as well as the global surgery research teams at the University of California, San Francisco, and the University of California, Los Angeles.

## Author Contributions

**Conceptualization:** Dennis J. Zheng, Catherine Juillard, Mary Margaret Ajiko, Rochelle A. Dicker.

**Data curation:** Patrick J. Sur.

**Formal analysis:** Dennis J. Zheng, Patrick J. Sur.

**Funding acquisition:** Rochelle A. Dicker.

**Investigation:** Mary Goretty Ariokot.

**Methodology:** Dennis J. Zheng, Patrick J. Sur, Catherine Juillard, Mary Margaret Ajiko, Rochelle A. Dicker.

**Project administration:** Dennis J. Zheng, Mary Goretty Ariokot, Mary Margaret Ajiko.

**Resources:** Mary Goretty Ariokot.

**Supervision:** Catherine Juillard, Mary Margaret Ajiko, Rochelle A. Dicker.

**Validation:** Dennis J. Zheng, Rochelle A. Dicker.

**Visualization:** Patrick J. Sur.

**Writing – original draft:** Dennis J. Zheng.

**Writing – review & editing:** Dennis J. Zheng, Patrick J. Sur, Mary Goretty Ariokot, Catherine Juillard, Mary Margaret Ajiko, Rochelle A. Dicker.

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
