## [Decision Letter · Decision Letter 0]

2 Nov 2020

PONE-D-20-26572

Epidemiology of Injured Patients in Rural Uganda: A Prospective Trauma Registry’s First 1000 Days

PLOS ONE

Dear Dr. Dicker,

Thank you for submitting your manuscript to PLOS ONE. After careful consideration, we feel that it has merit but does not fully meet PLOS ONE’s publication criteria as it currently stands. Therefore, we invite you to submit a revised version of the manuscript that addresses the points raised during the review process.

We look forward to receiving your revised manuscript.

Kind regards,

Zsolt J. Balogh, MD, PhD, FRACS

Academic Editor

PLOS ONE

Journal Requirements:

2. Please amend your current ethics statement to address the following concerns: Please explain why written consent was not obtained, how you recorded/documented participant consent, and if the ethics committees/IRBs approved this consent procedure.

3. Please include additional information regarding the survey or questionnaire used in the study and ensure that you have provided sufficient details that others could replicate the analyses. For instance, if you developed a questionnaire as part of this study and it is not under a copyright more restrictive than CC-BY, please include a copy, in both the original language and English, as Supporting Information. In addition, please describe the development and validation of this tool in further detail, including a thorough description of the FGDs held and the number of participants included in the pilot testing of this tool.

4. Please include your tables as part of your main manuscript and remove the individual files. Please note that supplementary tables (should remain/ be uploaded) as separate "supporting information" files.

Reviewers' comments:

Reviewer's Responses to Questions

**Comments to the Author**

1. Is the manuscript technically sound, and do the data support the conclusions?

Reviewer #1: Yes

Reviewer #2: Yes

2. Has the statistical analysis been performed appropriately and rigorously? 

Reviewer #1: Yes

Reviewer #2: N/A

3. Have the authors made all data underlying the findings in their manuscript fully available?

Reviewer #1: Yes

Reviewer #2: Yes

4. Is the manuscript presented in an intelligible fashion and written in standard English?

Reviewer #1: Yes

Reviewer #2: Yes

5. Review Comments to the Author

Reviewer #1: Dear Rochelle and team

You report the establishment of a trauma registry in a regional rural facility in eastern Uganda and detail the injury mechanisms, age and sex differentiation over time, defining the common injury patterns. This is laudable work. I have only some minor technical comments and some suggestions to slightly expand the discussion.

1) As per the recent SABRE guidelines I suggest where you refer to GENDER this is changed to SEX (the former is self-determined and the latter genetically fixed)

2) Some minor expansion of the discussion to include not only the other Ugandan reports on injury burden, but also the other countries around Uganda would be useful (recent work from Botswana and Tanzania published in WJS or AfJEM and from Malawi in various journals would be useful for comparison, not to mention the numerous papers detailing a very different injury pattern from South Africa

3) A discussion around the NSOAP recently completed for Uganda would be useful to contextualize the paper

4) A clear conclusion that is not stated strongly enough as a recommendation would be the establishment of a proper Emergency Department at the hospital.

A well written and timely paper on an important aspect of the 4th burden of disease in LMICs

Reviewer #2: The authors describe the initiation of a trauma registry in Uganda, after analysis of the first 1000 days. The authors are to be complimented in their effort to set up such a registry and attempt to improve prevention and treatment of trauma victims in their country. Nevertheless, several questions remain:

* I understand the reason why the KTS was used, it makes comparison with other international studies difficult. Furthermore, this score mainly contains physiological parameters, parameters which are known to be missing when not collected miticuleously (in the present study 48% missing data). And when they are collected, these parameters are severely influenced by the pre-hospital setting (distance/time, treatment given/not given etc). Is additional information available for (at least a part of) these patients, such as ISS, NISS? The advantage of these scores based on anatomy is they can be collected in hindsight.

* A statement is made that the registry influences trauma outcome. Do the authors have data on this statement? Can this be further elaborated?

* The number of patients included in the registry declines over time (each period, the number is cut in half). This might be one of the most important findings of this paper. Can there be reasons identified for this phenomenon? How can other centers learn from this experience?

6. PLOS authors have the option to publish the peer review history of their article (what does this mean?). If published, this will include your full peer review and any attached files.

Reviewer #1: No

Reviewer #2: No

---

## [Author Response · Author response to Decision Letter 0]

17 Dec 2020

RESPONSE TO REVIEWERS

The original comments by Reviewer 1 and Reviewer 2 are listed below, followed by our response to each comment:

Reviewer 1:

1. As per the recent SABRE guidelines I suggest where you refer to GENDER this is changed to SEX (the former is self-determined and the latter genetically fixed)

This has been corrected in each instance. 

2. Some minor expansion of the discussion to include not only the other Ugandan reports on injury burden, but also the other countries around Uganda would be useful (recent work from Botswana and Tanzania published in WJS or AfJEM and from Malawi in various journals would be useful for comparison, not to mention the numerous papers detailing a very different injury pattern from South Africa

Studies describing findings from other sub-Saharan African trauma registries were added to the discussion in order to better contextualize our work.

3. A discussion around the NSOAP recently completed for Uganda would be useful to contextualize the paper

We were unable to discover much about the current status of Uganda’s NSOAP in the literature, nor are our local collaborators involved in its development, but this will undoubtedly be a landmark step in the advancement of care for surgical patients in Uganda.

4. A clear conclusion that is not stated strongly enough as a recommendation would be the establishment of a proper Emergency Department at the hospital.

This is a valuable lesson learned from our registry implementation and has been better emphasized in our conclusion.

Reviewer 2:

1. I understand the reason why the KTS was used, it makes comparison with other international studies difficult. Furthermore, this score mainly contains physiological parameters, parameters which are known to be missing when not collected miticuleously (in the present study 48% missing data). And when they are collected, these parameters are severely influenced by the pre-hospital setting (distance/time, treatment given/not given etc). Is additional information available for (at least a part of) these patients, such as ISS, NISS? The advantage of these scores based on anatomy is they can be collected in hindsight.

Many trade-offs were made in the development of the registry’s instrument, in particular balancing the desire for detailed data collection while maintaining the feasibility of the project. The hospital’s workflow demands and lack of widespread clinical expertise prevented us from reliably establishing an Injury Severity Score for the patients in the registry, whether prospectively or retrospectively. Previously published work from our group has demonstrated the significant predictive value for hospital mortality of physiologic scoring systems such as the Kampala Trauma Score (KTS). We felt this approach would maximize practicality and allow for adequate comparisons with other trauma registries located in low- & middle-income settings. The inability to reliably record all of the elements making up the KTS is a multifactorial problem and represents a focus of ongoing quality improvement measures, in part driven by the trauma registry.

2. A statement is made that the registry influences trauma outcome. Do the authors have data on this statement? Can this be further elaborated?

In our discussion we have described in further detail the possible downstream effects of trauma registry implementation. Given the extensive education efforts associated with the establishment and maintenance of the registry, it is not inconceivable that clinical providers could have altered their behaviors to emphasize collection of vital signs, for example. While the Hawthorne effect may influence the practice of clinical care in this way, we would hesitate to say that it may affect patient outcomes.

3. The number of patients included in the registry declines over time (each period, the number is cut in half). This might be one of the most important findings of this paper. Can there be reasons identified for this phenomenon? How can other centers learn from this experience?

In our revised discussion we describe in further detail the varied factors contributing to the decline in registry patients over time. They encompass a range of barriers to success in global health, many of which have fatally afflicted trauma registries around the world. Although our data collection is not as robust as it once was, we have successfully adapted our approach to meet the challenge of sustainability, and we consider our currently operational database to serve as an example of the immense capabilities of a locally-run trauma registry.

---

## [Decision Letter · Decision Letter 1]

8 Jan 2021

Epidemiology of Injured Patients in Rural Uganda: A Prospective Trauma Registry’s First 1000 Days

PONE-D-20-26572R1

Dear Dr. Dicker,

We’re pleased to inform you that your manuscript has been judged scientifically suitable for publication and will be formally accepted for publication once it meets all outstanding technical requirements.

Kind regards,

Zsolt J. Balogh, MD, PhD, FRACS

Academic Editor

PLOS ONE

Additional Editor Comments (optional):

Reviewers' comments:

Reviewer's Responses to Questions

**Comments to the Author**

1. If the authors have adequately addressed your comments raised in a previous round of review and you feel that this manuscript is now acceptable for publication, you may indicate that here to bypass the “Comments to the Author” section, enter your conflict of interest statement in the “Confidential to Editor” section, and submit your "Accept" recommendation.

Reviewer #1: All comments have been addressed

2. Is the manuscript technically sound, and do the data support the conclusions?

Reviewer #1: Yes

3. Has the statistical analysis been performed appropriately and rigorously? 

Reviewer #1: Yes

4. Have the authors made all data underlying the findings in their manuscript fully available?

Reviewer #1: Yes

5. Is the manuscript presented in an intelligible fashion and written in standard English?

Reviewer #1: Yes

6. Review Comments to the Author

Reviewer #1: (No Response)

7. PLOS authors have the option to publish the peer review history of their article (what does this mean?). If published, this will include your full peer review and any attached files.

Reviewer #1: No

---

## [Editor Report · Acceptance letter]

13 Jan 2021

PONE-D-20-26572R1 

Epidemiology of Injured Patients in Rural Uganda: A Prospective Trauma Registry’s First 1000 Days 

Dear Dr. Dicker:

I'm pleased to inform you that your manuscript has been deemed suitable for publication in PLOS ONE. Congratulations! Your manuscript is now with our production department. 

Kind regards, 

on behalf of

Dr. Zsolt J. Balogh 

Academic Editor

PLOS ONE